# Feasibility of an Intelligent Algorithm Based on an Assist-as-Needed Controller for a Robot-Aided Gait Trainer (Lokomat) in Neurological Disorders: A Longitudinal Pilot Study

**DOI:** 10.3390/brainsci13040612

**Published:** 2023-04-04

**Authors:** Caroline Laszlo, Daniele Munari, Serena Maggioni, Deborah Knechtle, Peter Wolf, Dino De Bon

**Affiliations:** 1Sensory-Motor Systems (SMS) Lab, ETH Zurich, 8006 Zurich, Switzerland; caro.laszlo@swissonline.ch; 2Hocoma AG, 8604 Volketswil, Switzerland; 3Revigo, Rehaklinik Zihlschlacht AG, 8604 Volketswil, Switzerland

**Keywords:** neurological rehabilitation, robotics, assist-as-needed, walking, functional assessment, Lokomat

## Abstract

Most robotic gait assisted devices are designed to provide constant assistance during the training without taking into account each patient’s functional ability. The Lokomat offers an assist-as-needed control via the integrated exercise “Adaptive Gait Support” (AGS), which adapts the robotic support based on the patient’s abilities. The aims of this study were to examine the feasibility and characteristics of the AGS during long-term application. Ten patients suffering from neurological diseases underwent an 8-week Lokomat training with the AGS. They additionally performed conventional walking tests and a robotic force measurement. The difference between robotic support during adaptive and conventional training and the relationship between the robotic assessment and the conventional walking and force tests were examined. The results show that AGS is feasible during long-term application in a heterogeneous population. The support during AGS training in most of the gait phases was significantly lower than during conventional Lokomat training. A relationship between the robotic support level determined by the AGS and conventional walking tests was revealed. Moreover, combining the isometric force data and AGS data could divide patients into clusters, based on their ability to generate high forces and their level of motor control. AGS shows a high potential in assessing patients’ walking ability, as well as in providing challenging training, e.g., by automatically adjusting the robotic support throughout the whole gait cycle and enabling training at lower robotic support.

## 1. Introduction

Independent walking is among the most desired goals of patients after a neurological injury [1,2]. Being able to move in the community facilitates activity and participation in daily life [3]. Patients suffering from neurological diseases should start rehabilitation as soon as possible and should be provided with intensive training to achieve effective therapy results [4]. The intensity of therapy delivered can be increased by introducing robotic devices [5,6,7]. In order to elicit the maximal voluntary contribution of the patient, robotic devices should reduce their support to a minimum while enabling safe training. The lower the robotic support, the more patients are motivated to contribute to the movement, thereby potentially increasing their active participation [8,9]. In exoskeleton robots, control strategies provide support based generally on position control or on force control. In position control, the robotic joints are programmed to follow predefined trajectories; the impedance of the controller regulates how compliant they are to deviations. The relationship between deviation and applied force is regulated by the impedance, which can be fixed or adaptable (manually or automatically) [10,11]. High impedance leads to a very constrained gait pattern, while low impedance increases compliance while reducing physical support to the movement. With force control, no pre-defined trajectory exists, and the control behavior is defined through reactions to specific actions by the patients, defined by the events of gait as a cyclic and rhythmic process. This may result in more freedom in imposing one’s specific gait pattern but may also lead to a higher risk of abnormal trajectories, especially if the subject’s impairment is severe [8,11,12,13].

However, most robotic devices do not automatically adapt to patients’ abilities and rely only on therapists’ judgement to manually adjust the robotic support. This can lead to a robotic support which is not optimal for the patient and a failure to challenge the patient appropriately [10,11]. In order to address these problems, an assist-as-needed robotic controller can be used. The level of robotic assistance needed is modulated based on a continuous assessment of the individual patient’s ability. Efforts have been made to develop such assist-as-needed (AAN) controllers for robotic gait rehabilitation devices [9,14,15,16]. Until now, though, these AAN controllers have been mostly limited to research applications and have not been implemented in clinical practice, i.e., in commercially widely used devices.

The Lokomat is a treadmill-based robotic exoskeleton which has been widely used in clinical practice worldwide since the early 2000s. The effects of Lokomat therapy have been extensively studied in several patient populations (stroke [17,18], spinal cord injury (SCI) [19], and multiple sclerosis [20], to name a few). In this study, an intelligent algorithm called adaptive gait support (AGS), recently implemented in Lokomat^®^Pro (Sensation exercise package, Hocoma AG, Volketswil, Switzerland) [21,22], was tested. AGS has been implemented as a training exercise accessible from the Lokomat user interface. AGS automatically and continuously adjusts the robotic support of the Lokomat orthosis based on the patient’s ability to follow a physiological gait pattern, which should be related to the patient’s level of impairment [22].

The feasibility of an initial research version of AGS and its assessment properties have been studied in two sessions with people after SCI [22]. In this study, fifteen participants with walking impairments and twelve able-bodied persons used AGS during two Lokomat training sessions. The results showed that AGS can be used to quantify the support required by a patient while performing robotic gait training and is feasible in people with very different levels of gait impairment [22]. However, no studies on the long-term application of AGS, i.e., integrating AGS into the patient’s exercise plan over several weeks, have been performed yet in pathologies other than SCI.

The primary aim of the current study was to examine the feasibility of a commercially available AGS with chronic neurological patients with different diagnosis in real-world clinical practice during an 8-week training period. Our hypothesis was that AGS is feasible for patients of diverse neurological diseases. The secondary aim of the study was to explore the data obtained from AGS, in order to observe its performance as an AAN training tool and as an assessment tool for walking ability. Within-subject changes of robotic support over the course of the 8-week training were monitored. We also hypothesized that the average robotic support automatically determined during AGS training would be reduced compared to conventional Lokomat training, where the support is manually adapted by the therapist and cannot be targeted to specific phases of the gait cycle. The validity of AGS as an assessment tool (i.e., relationship to conventional walking assessments and to an isometric force assessment) and reliability were also examined.

## 2. Materials and Methods

### 2.1. Trial Design

A longitudinal pilot study was conducted between October 2020 and February 2021. All study participants were recruited at the outpatient rehabilitation clinic Revigo in Volketswil (Switzerland), where they already attended robotic therapy.

All patients gave their informed, written consent to participate in the study, which was carried out according to the Declaration of Helsinki and approved by the ethics commission of ETH Zurich (EK 2020-N-139).

### 2.2. Participants

Patients with a chronic (>6 months) central nervous, peripheral neurogenic, spinal, muscular, or bone-related disorder with limited walking abilities were included in the study. The inclusion criteria were (i) age > 18 years and (ii) having passed the Montreal Cognitive Assessment (Version 8.1, ≥26/30 points) (MoCA) [23], which was used to assess the patients’ cognitive abilities. As the aim of the study was to assess the feasibility of AGS in patients with gait impairments regardless of the underlying pathology, we did not limit the inclusion criteria to a single pathology. The exclusion criteria were (i) existence of a contraindication outlined in the manufacturer’s manual (Hocoma AG, Volketswil, Switzerland, www.hocoma.com, accessed on 1 March 2023), (ii) inability or unwillingness to give written consent, and (iii) inability or unwillingness to follow the study protocol. Ten patients were finally included (see Table 1).

### 2.3. Experimental Apparatus and Control

All training and assessment sessions were conducted on a LokomatPro V6 FreeD (Hocoma AG, Volketswil, Switzerland) in the outpatient rehabilitation center Revigo in Volketswil (Switzerland). The Lokomat is a treadmill-based robotic exoskeleton with actuated hip and knee joints and a dynamic body-weight-support system that supports the patient through a harness [24]. The orthosis is programmed to follow a predefined gait pattern with an impedance control strategy. In standard therapy, the gait pattern and the mechanical impedance of the robotic hip and knee joints (called “guidance force” in the Lokomat software) are manually adjusted by the therapist through a user interface. The guidance force is constant throughout the gait cycle.

AGS is a newly developed software modality based on an assist-as-needed controller, available in the LokomatPro Sensation exercise package [21,22]: AGS adjusts the joints’ mechanical impedance automatically and continuously throughout the gait cycle. The patients are requested to follow a reference foot trajectory on the screen. The controller automatically adapts the mechanical impedance (i.e., normalized stiffness *K* and damping *B*) of the hip and knee joints of the Lokomat based on the deviation *e* between a reference and the actual foot trajectory. In order to optimally support each gait phase, 30 windows *w* per step are implemented and the impedance is adapted separately in each window [22]. At every gait cycle *s* and at each window *w*, *K* and *B* were adapted based on an error metric *f(e)_s,w_* (Equations (1) and (2)). The error metric allows deviations from the reference trajectory within predefined deadbands around the hip and knee reference trajectories, which were set equally for all participants. A set of gains *g* was implemented in order to react quickly in case of large errors. Additionally, a forgetting factor *g* < 1 limited the excessive reliance on the assistance provided by the controller. More details on the technical implementation of the controller have been reported elsewhere [22].
(1)$$K_{s+1,w}=\gamma_{1}\cdot K_{s,w}+g_{1}\cdot f_{1}{\left(e\right)}_{s,w}$$
(2)$$B_{s+1,w}=\gamma_{2}\cdot B_{s,w}+g_{2}\cdot f_{2}{\left(\dot{e}\right)}_{s,w}$$

Initially, the normalized mechanical impedance of the Lokomat was set to 100% (maximum *K* and *B* achievable by the controller) and was then gradually adapted based on the patient’s kinematic error in the different phases of the gait cycle. Small kinematic deviations from the reference lead to a decrease in *K* and *B*, until a minimum threshold set by the therapist for safety is reached (set to 4% in this study). Large kinematic deviations lead instead to an increase in *K* and *B*.

The software was programmed so that, during AGS training, the adaptive controller was applied on one leg only, whereas the other leg was constantly supported with 100% assistance. This helps the patient to concentrate on the gait cycle of one leg, which is displayed as visual feedback on the screen (see Figure 1).

The patient could see her/his last two gait cycles of the active leg displayed and also received feedback (parts of the gait cycle with small errors were displayed in green and parts with large kinematic deviations in red) on the actual robotic support in the different phases of the gait cycle.

The AGS exercise could be accessed via the therapy plan of the Lokomat (software 6.5.8, Sensation exercise package).

### 2.4. Protocol

Prior to the study start, all participants completed two familiarization sessions with AGS to define the individual Lokomat settings which included the Lokomat’s hip and knee trajectory offset, range of motion (ROM), lateral pelvis movement, and gait speed. These individual settings were held constant for the whole study period. No fixed value for the body-weight support (BWS) provided by the Lokomat was defined, although the therapist was instructed to ensure that the patient trained within the dynamic range of the BWS system (i.e., where the BWS system ensures a constant unloading throughout the gait cycle). The amount of BWS was set to guarantee an upright body position and to avoid knee buckling during each stance, simultaneously ensuring challenging training in a safe environment. The therapist was also asked to avoid large session-to-session changes in BWS. After the two familiarization sessions, the participants trained with the Lokomat for eight weeks, conducting AGS at least once a week.

AGS was always conducted at the beginning of the training session after an initial warm up on the Lokomat (e.g., to reduce spasticity). The patient then trained with AGS for 10 min: during minutes 1–5, the support was adapted for the left leg, and during minutes 6–10 for the right leg. Afterwards, the patients continued with their usual training programs. During this conventional Lokomat training (i.e., without AGS), the therapist manually selected the level of hip and knee robotic support (using the “guidance force” setting), which he/she thought was the optimal support level for this patient.

In addition to the AGS training sessions, the participants completed three assessment sessions, conducted in weeks 1, 5, and 8. These sessions included (i) a 10 meter-walk test (10MWT) (2 runs, separated by a break) [25], (ii) a timed up and go test (TUG) (2 runs, separated by a break) [26], (iii) a functional ambulatory category (FAC) [27], (iv) a robotic assessment to measure isometric joint torque called L-FORCE [28], and (v) questionnaires to assess health-related quality of life and self-reported health status: the European Quality of Life 5-Dimension 3-Level questionnaire (EQ-5D-3L) [29] and the World Health Organization Disability Assessment Schedule 2.0 (WHODAS 2.0., 36-Item Version) [30]. At the end of the study, a questionnaire to assess the acceptance of AGS was conducted. The questionnaire was based on the technology acceptance model (TAM) [31,32]. We documented any adverse events that occurred during training and assessment sessions.

### 2.5. Data Analysis

Data processing was performed using MATLAB^®^ (Version R2020a, The MathWorks Inc., Natick, MA, USA). The recorded signals of the Lokomat provided continuous data collected every 4 ms and included normalized stiffness *K* and damping *B* (from 0 to 1) data for the knee and hip of both legs, in addition to hip and knee reference and actual angular trajectories.

#### 2.5.1. Feasibility

The acceptability of the AGS in patients was measured with the TAM questionnaire. Five determinants were identified as related to technology acceptance. The determinants were (i) enjoyment, (ii) anticipated use, (iii) perceived usefulness, (iv) perceived ease of use, and (v) overall evaluation [31]. A Likert scale from 1 to 7 (1 corresponds to ”strongly agree” and 7 to “strongly disagree”) was used. For every answer, the scores were averaged among patients.

#### 2.5.2. AGS as a Training Tool

During training with AGS, the mean of the normalized stiffness for the knee and the hip were calculated separately for the left and right leg. The first minute was excluded from the calculation as the normalized stiffness adapted according to Equation (1) required some time to decrease (exponential decay) to a point where the patient was actually able to influence the gait trajectory. The normalized stiffness of the knee and the hip were also calculated during the entire conventional Lokomat training and then averaged over both legs.

#### 2.5.3. AGS as an Assessment Tool

The normalized stiffness and damping variables were further subdivided and analyzed according to the gait phase. Each gait step was divided into different phases based on the reference trajectories of hip and knee, according to the literature [33,34] and adjusted based on experience in a previous study [22]. The subphases of interest were, in the swing phase, the initial swing (IS, 10.5% gait cycle), mid swing (MS, 14% gait cycle), and terminal swing (TS, 13% gait cycle) phases. The stance phase was not analyzed, since the tracking of the reference trajectory during each stance is highly affected by the amount of BWS and by the contact with the treadmill. Subsequently, the average stiffness and damping of every subphase and of every step was calculated for the knee, as well as the hip, of both legs. This resulted in 12 data points for every step: the stiffness/damping of the knee was subdivided into the three subphases and the stiffness/damping of the hip was subdivided into the three subphases, calculated for the left and right leg individually.

As a next step, an exponential decay curve was fitted to the data of each subphase. Either the value of the plateau of the curve or, in case of no existing plateau, the mean of the last 4 min of the condition was used in the analysis. This approach resulted in 12 different values of stiffness/damping for each leg and per training session.

For the comparison of the AGS variables with the gait assessments, the calculated values were averaged over the left and right leg for every individual assessment session, which resulted in 6 final stiffness values (knee stiffness of all 3 gait phases and hip stiffness of all 3 gait phases averaged over the left and right leg) and 6 final damping values (knee damping of all 3 gait phases and hip damping of all 3 gait phases averaged over the left and right leg).

The results of the 10MWT were converted into m/s. The average of both runs for the 10MWT, as well as for the TUG, was then calculated for each assessment session.

The L-FORCE data were normalized to the patient’s body weight, and subsequently the mean over all 4 different movements (flexion and extension for hip and knee) was calculated for both legs individually.

### 2.6. Statistical Analysis

Statistical analysis was performed using R (Version 4.0.3, R Foundation for Statistical Computing, Vienna, Austria). Descriptive statistics were calculated using MATLAB^®^ (Version R2020a, The MathWorks Inc., Natick, MA, USA).

#### 2.6.1. AGS as a Training Tool

In order to test whether the difference of the average robotic support level automatically determined by the AGS algorithm and that manually adjusted by the therapist during conventional Lokomat training was significant, it was tested whether the data were normally distributed, and a paired *t*-test was conducted. The paired *t*-test was selected because differences in means were under investigation. The normalized stiffness averaged among the knee and the hip was used as a variable describing the robotic support level.

#### 2.6.2. AGS as an Assessment Tool—Validity

In order to identify the AGS variable(s) which can best explain the variation in walking ability over all patients (measured by walking assessments), aggregated data (data of every participant averaged across the three assessments) of all 12 variables (“stiffness knee initial swing”, “stiffness knee mid swing”, “stiffness knee terminal swing”, “stiffness hip initial swing”, “stiffness hip mid swing”, “Stiffness hip terminal swing”, “damping knee initial swing”, “damping knee mid swing”, “damping knee terminal swing”, “damping hip initial swing”, “damping hip mid swing”, and “damping hip terminal swing”) were used to fit a linear regression model—used in an explorative manner as a variable selection method. Only patients who could perform the walking tests were included in this analysis. The forward selection method was applied to generate this model. In every step, only the variable with the lowest *p*-value was included in the model if the value was <0.05. This procedure was continued until no additional significant variable could be included in the model. The variable selection was conducted with two different models: one with 10MWT and one with TUG as the dependent variable. In order to confirm the results of the variable selection method applied to our dataset, which contained variables prone to collinearity, we also applied a variable selection method based on Elastic Net [35], using the MATLAB function lasso (alpha 0.75 for Elastic Net, 3-fold cross-validation). Since cross-validation is used to determine the best regularization coefficient, lasso returns different results every time. In order to check whether the results were consistent, lasso was run 100 times. We then looked at the selection frequency of each variable.

#### 2.6.3. Longitudinal Within-Patient Data

In order to explore the change in AGS data within patients over the course of the training period, given the exploratory characteristics of the study and the limited number of subjects, we decided not to perform a statistical analysis but rather to present the data graphically. The variable chosen in Section 2.6.2 was plotted against the walking tests 10MWT and TUG (in this case, all three data points from each patient were considered). Moreover, the same variable was plotted for the left and right leg, for every patient, and for every Lokomat session where AGS was used, in order to show the progression of AGS data over time. Additionally, the change from the first to the last training session in the variable chosen in Section 2.6.2 was calculated for the average of the left and the right leg for every patient.

#### 2.6.4. Relationship between L-FORCE and AGS

For the analysis of the relationship between L-FORCE measurements and AGS data, a linear regression model was fitted with the aggregated data of the normalized L-FORCE and AGS of each leg. To analyze the effects over all patients, the L-FORCE and AGS variables were averaged for every participant over all three assessment sessions. One linear regression model was fitted for the right leg and another model for the left leg, and subsequently it was analyzed whether the correlation between the normalized L-FORCE of one leg and the AGS data of the same leg was significant.

#### 2.6.5. Reliability

In order to assess the reliability of the assessment, within each patient, the data extracted from the two AGS sessions that were closest in time in the second half of the study period were included. The training sessions were performed by the same therapist, hence intra-rater reliability was examined. The variable chosen from the forward selection method (Section 2.6.2) was tested for reliability. Relative reliability was evaluated using Spearman’s rank correlation coefficient (Rho). Absolute reliability was assessed using the Bland–Altman plot with 95% limits of agreements (LOAs). The limits of agreements (LOAs) indicate the range where, for a new person from the studied population, the difference between any two tests lies within a 95% probability. Changes between two measurements are considered significant only if they fall outside of the LOAs [36]. LOAs are expressed as a percentage of the mean of the data.

All models were tested for their model assumptions and *p*-values < 0.05 were considered statistically significant.

## 3. Results

### 3.1. Participants

The patient characteristics were highly heterogenous (see Table 1). The patients did not only differ in type of diagnosis but also in age, time since diagnosis, BMI, and FAC score.

An average of 9.9 (min 8, max 16) AGS training sessions were conducted per patient within the 8 weeks of the study. Three patients were non-ambulatory at the time of the study and could not perform the walking tests. One patient did not perform the walking tests for organizational reasons.

Patient one was suffering from bowel cancer and underwent chemotherapy during the study period. Chemotherapy was not defined within the exclusion criteria and over the whole study period, there was no necessity to stop Lokomat training or exclude the patient from the study. The primary diagnosis of patient three was multiple sclerosis. Epidural abscesses which needed to be removed by surgery were the reason for walking inability. The impairment of patient five was probably induced by a viral infection in his childhood. This patient wanted to interrupt his therapy due to the COVID-19 pandemic. During a period of 4 weeks, he conducted 10 training sessions. It was decided to include the patient’s data in the analysis because he attended more than 8 training sessions. His data only includes 2 assessment sessions. Patient six did not pass the MoCA. The main reason was her spatial neglect, which made it impossible to solve two tasks of the MoCA. Due to her higher education (patient was attending a master’s degree during the study period), it was decided to include the patient in the study. Due to sickness, this patient had a training interruption of 12 days between sessions three and four. In addition, due to sickness and Christmas holidays, patient eight had a training interruption of 18 days between training sessions seven and eight (assessment three). The number of conducted training sessions with the AGS software is displayed in Table 2.

A summary of all tests which were conducted by the participants is displayed in Table 3.

### 3.2. Primary Aim—Feasibility

Training with AGS was feasible for all participants. No adverse events occurred, and patients did not report fatigue during the AGS exercise. AGS had a high acceptability among patients, revealed by the answers of the TAM questionnaire (see Figure 2). The mean score of the answers for every question was in the lower half of the scale, indicating a positive outcome. The patients found AGS easy to use and useful, and 8/10 patients would use AGS again.

### 3.3. Secondary Aims

#### 3.3.1. AGS as a Training Tool

The comparison of the robotic support of the Lokomat during AGS (automatically set by the controller) and conventional training (manually set by the therapist) resulted in a significantly lower robotic support during AGS training than conventional Lokomat training for all patients (*p*-value = 6.74 × 10^−8^). The robotic support during AGS training was adapted throughout every gait cycle by the controller and was not constant throughout the gait cycle—as it is during conventional Lokomat training (see Figure 3).

#### 3.3.2. AGS as an Assessment Tool: Validity

The analysis of the most suitable variable to explain the variation in walking ability over all patients (i.e., with a cross-sectional approach) resulted in two models, one model with 10MWT and one with TUG as the dependent variable. In both models, the variable knee stiffness initial swing phase was chosen with the forward selection method as the best variable to explain the variation in walking ability (see Figure 4). Only one variable was chosen because in step two of the stepwise variable selection, no additional variable could be selected for the model (Table 4 and Table 5).

The variable knee stiffness initial swing phase was also the most frequently selected by the elastic net method.

Six data points are shown because only six patients were able to conduct the overground walking tests. The coefficient in the 10MWT model is negative, which indicates that the higher the support required in the initial swing phase of the knee, the lower the speed (for every percent additional knee stiffness initial swing: −0.04 m/s in 10MWT). The coefficient in the TUG model is positive, which means that higher support was related to more time needed to conduct the test (for every percent additional knee stiffness initial swing: +3.38 s in TUG).

#### 3.3.3. Longitudinal Within-Patient Data

The variable selected in Section 3.3.2 was plotted for every patient and every AGS session against the walking tests results (10MWT and TUG) (Figure 5).

In certain patients, K KNEE IS (%) decreased over time (e.g., P2), whereas in other patients it increased (e.g., P8). P5 only has two data points (interruption of training because of the COVID-19 pandemic).

The stiffness determined automatically by AGS for each patient during the 8-week training period (see Figure 6) shows a high variability between and within patients (especially in P2, P3, and P4). Differences between legs were also observed in several patients (P3, P4, and P9). Certain trends of decreasing stiffness over time could be observed in P2, P3, and P4.

Figure 6 shows that certain patients could improve their values (i.e., decrease K KNEE IS), while others stayed on the same level or even worsened. Table 6 shows the percentage of change between the first and last AGS session. It can also be seen that there is a lot of variability in the AGS data of particular patients between sessions. P2 (second left on top row) and P3 (third left on top row) were the patients with the most evident improvement in the AGS data, which meant that they needed a lower amount of robotic support. This was also observable in practice: P3 was able to exercise in an upright position and do certain steps (however, this patient was not able to complete the 10MWT). In P2, it was observed that his gait was much smoother and more physiologically correct, which could indicate that his motor control had improved. P2 shows a high variability in the AGS data. The patient suffered from an inflammatory central nervous disorder. The AGS data reflected that on days with increased inflammation, motor control was more difficult.

#### 3.3.4. Relationship between L-FORCE and AGS

The regression model analysis which investigated the relationship between the normalized L-FORCE measures and AGS data (K KNEE IS (%)) of every leg individually showed no linear relationship between the variables. It was not possible to predict normalized L-FORCE measures with AGS data with p < 0.05, and therefore no model could be created. However, when plotting this variable versus L-FORCE measures (Figure 7), patients could be divided into three clusters: (i) low L-FORCE and low AGS data, (ii) low L-FORCE and high AGS data, and (iii) high L-FORCE and low AGS data. All patients who were able to walk had a high maximum isometric force measured by the L-FORCE and they reached a low knee stiffness at initial swing, whereas the non-ambulatory patients of the study cohort all showed low force values, though some reached the same low knee stiffness as the ambulatory patients.

#### 3.3.5. Reliability

The relative reliability of the knee stiffness of the initial swing phase resulted in a Spearman’s coefficient (Rho) of 93%. The absolute reliability analysis of the knee stiffness of the initial swing phase using the Bland–Altman plot produced LOAs of 17.0%. Therefore, any difference in the knee stiffness of the initial swing phase between two measures lower than 17.0% cannot be considered a significant change.

## 4. Discussion

### 4.1. Feasibility of AGS

The primary aim of this study was to examine the feasibility of the AGS exercise. All ten patients regardless of their diagnosis or level of ambulation were able to train with AGS. The findings indicate that AGS training is feasible and safe in a long-term application in clinical practice. Due to the characteristics of the study site, which was an outpatient training center, only outpatients in the chronic phase of their disease were included in the study. Given that the range of impairment was very high (FAC levels ranged from 0 to 5), it can be assumed that AGS would also be feasible with patients in a (sub-)acute phase, provided that they are not cognitively impaired.

These results were also confirmed by the positive answers to the TAM questionnaire. Evidence in the literature also indicates that patient-cooperative training with visual feedback is well accepted by patients, improves their active participation, and means that they are able to perform these kind of tasks [9,37].

### 4.2. Adaptive Gait Support as a Training Tool

This study also aimed to investigate the characteristics of AGS as a training tool. The difference between the robotic support determined automatically by the AGS and the support manually determined by the therapist during conventional Lokomat training was also examined. Additionally, the behavior of the support throughout the gait cycle was examined.

In almost every gait phase, all patients reached significantly lower levels of robotic support during AGS training than during conventional Lokomat training, where the support was set manually by the therapist. Thus, regardless of the diagnosis or level of impairment of the patient, the adaptive controller set lower robotic support levels during AGS training than the therapists set. It has to be noted, however, that during conventional Lokomat training the therapist was only asked to set the robotic support to a level which he/she thought was best for the patient and not to lower the support to a minimum, while during AGS, the controller determined the most suitable robotic support for each patient, and for each gait subphase. Despite recommending that the patient is challenged as much as possible, it is not unusual for therapists to maintain a high robotic support (“guidance force”) during clinical practice. The therapists often prefer to be on the safe side and prevent the Lokomat from stopping (if the robotic support is set too low, kinematic deviations may trigger the safety mechanisms). In addition, due to the missing physical interaction between patient and therapist, the therapist can have difficulty determining how active the patient in the machine is. One reason for the high reduction in guidance force achieved in this study could be that AGS allows a higher support in the gait phases where it is more needed, while a low support can be reached in others. When the therapists manually set the guidance force in the standard Lokomat controller, they can only lower it as far as the weakest phases of the patient’s gait cycle allow. By calculating the kinematic deviation between the actual and reference foot trajectory in 30 different windows of the gait cycle, the AGS controller supports the therapist by finding the performance limit of the patient in every phase of the gait cycle, thereby providing targeted assistance throughout the gait cycle. Moreover, as the AGS software adapts the robotic support at every session, it can address the changing needs and improvement of patients, who are expected to improve during the course of rehabilitation, thereby requiring less and less support from the machine.

In the literature, it has been shown that robotic devices may lead to slacking (i.e., the patient’s effort is decreased during training because the robotic device provides too much assistance) [10] which could potentially lead to a decreased recovery. During training with AGS, the robotic support can be lowered automatically, which may prevent the above-mentioned issues. However, further research is needed to test whether AGS can prevent slacking, as well as to examine whether AGS is superior compared to other robot-aided gait trainers and controllers in terms of optimal challenge of the patients.

When investigating the applied robotic support values on the Lokomat reported in other studies, we see that the minimum support values that could be reached in most studies were generally higher compared to the support levels reached in this study using AGS. Additionally, it was stated that attempting to lower the support to a minimum is especially challenging [38,39,40]. An explanation for this discrepancy could be the different patient population or the fact that the other studies could not appropriately target the different gait phases, maintaining a constant level of support throughout the gait cycle. However, Chang et al. and Mayr et al. lowered the support even more than in the current study but only included unilateral stroke patients, and it is not clear how many patients reached the minimal value [41,42].

### 4.3. Adaptive Gait Support as an Assessment Tool

A further aim of the study was to investigate the performance of AGS as an assessment tool by comparing its data to conventional walking tests and isometric force assessments and calculating the reliability of the assessment.

The analysis of the variation in walking ability explained by the AGS data resulted in a significant relationship between the measures of clinically established walking tests (10MWT and TUG) and AGS data (represented by robotic support at initial swing) focusing on the overall patient effect. The choice of the initial swing phase as a predictor of clinically established walking tests is in line with what is reported in the literature about this subphase—namely that the initial swing phase is a critical phase of the gait cycle [43]. The performance at the initial swing phase is particularly influenced by an appropriate push-off, which is one of the main determinants of gait speed and is very often impaired in neurological patients. During ankle push-off, most of the power during the gait cycle is generated [33].

Even though the results of this study showed no linear relationship between the isometric force (L-FORCE) and AGS data, we could hypothesize that the two measures reflect two separate components of walking ability: motor control (i.e., the ability to modulate the force) and the ability to generate high forces, as also suggested by Maggioni et al. [22]. The two measurements together can subdivide the patients into clusters (e.g., showing high or low isometric force or showing good or poor motor control). A clear separation between ambulatory patients and non-ambulatory patients, based on the maximum force the patients were able to achieve, could be made (see Figure 7). However, within the non-ambulatory patients, some reached low knee stiffness values in the AGS: this could be an index of a partially regained motor control with a persistent weakness. Research suggests that in order to regain walking ability, not only force [44] but also control is needed [5]. In Maggioni et al. [22], AGS and L-FORCE measures together increased the explained variance in a model with timed walking tests, suggesting that not only motor control (as measured by AGS) but also force (as measured by L-FORCE) was needed to explain the variance in walking ability expressed in a clinical test such as the 10MWT. Therefore, these two measures together could be an indicator of the patient’s progress from a non-ambulatory to an ambulatory stage and could be an additional tool for the therapist to evaluate how far the patient is from walking again. This possibility would be particularly relevant for patients with severe walking impairments, who have few other options for objective instrumented assessments of lower limb impairment and rely only on ordinal, coarse scores, such as the FAC.

When observing the progress of robotic support within the same patient over the 8-week training period, we did not observe trends in decreasing robotic support: this may be because patients were chronic, so little improvement in walking ability was expected, and therefore nothing could be captured by AGS. We also noticed a high variability between sessions. This may be for several reasons: AGS requires the patient to follow a trajectory displayed on the screen, a task that requires concentration and proper visuomotor coordination; therefore, distractions may influence the results. Moreover, patients had fluctuations in their health status due to symptoms such as inflammation and spasticity, also resulting in changes in performance during training.

The reliability analysis resulted in a good relative (using classification suggested in the literature [45]) but poor absolute reliability. This absolute reliability of the assessment should be improved before AGS can be used as an assessment in practice. The data which were considered for this analysis were one week apart for some patients, which may have led to increased variability due to patients’ differing daily health condition. In a future study, the reliability should be tested in sessions within one day or on two consecutive days, which may lead to an improved absolute reliability.

The AGS exercise can be integrated into the training procedure. To our knowledge, no commercially available robot-aided walking assessment exists, which is feasible in everyday practice without requiring additional time above that of the therapy.

Since AGS can be used with non-ambulatory patients in the Lokomat, before they can even perform a walking test (e.g., 10MWT), AGS may be able to assess the progress from the non-ambulatory to the ambulatory phase. An assessment for non-ambulatory patients is highly needed because, in terms of functional improvement, there are no feasible and valid performance-based measures of how close non-ambulatory patients are to walking again. Most clinical assessments are too crude to reveal functional changes [46] and lack the sensitivity to detect small changes in the rehabilitation process. A quantitative and responsive walking assessment for non-ambulatory patients might help to keep the patients motivated, as well as support the therapists in making data-driven decisions on the training program. However, the concurrent validity of such an assessment cannot be proven against established walking tests such as those used in this study (10MWT, TUG), as non-ambulatory patients would not be able to perform them.

### 4.4. Future Directions

Including participants in the (sub-)acute stages would be reasonable to detect larger changes in clinical scores and therefore a potentially significant effect within the patient. A clinical assessment must demonstrate a high responsiveness in order to be useful in clinical practice, and this aspect needs to be further improved and studied in future research on AGS.

The AGS assessment could be used to measure the asymmetry of the paretic and non-paretic leg by calculating the ratio of the legs. This ratio could be used as a helpful feature in addition to the L-FORCE data, which has been shown to be able to detect significant differences between the paretic and non-paretic leg [47]. Thus, the symmetry of the legs in terms of strength and control could be assessed.

No previous investigations have assessed the application of AGS in practice in a longitudinal study. This is the first study in this domain. These results support additional large-scale studies, where grouping the patients by diagnosis could provide more insights on the recovery trajectories of specific diseases. Further investigations should focus on the training effect of AGS and on improving the psychometric properties of AGS as an assessment tool, to enable it to capture the rehabilitation progress of non-ambulatory patients and therefore be used as a valid and reliable assessment tool. In principle, AGS as a training tool demonstrates beneficial features, such as requiring the patient to contribute actively to the conducted movements by lowering the robotic support in phases where less support was needed. These features could possibly contribute to more intensive and effective training. Additional studies are needed to provide evidence for this.

### 4.5. Limitations

This pilot trial has several limitations that restrict the strength of its conclusions. First, the sample size was small, which made it difficult for some parts of the analysis to achieve significant results. In addition, performing subgroup analysis based on patients’ diagnosis was not possible with this sample size. The diverse patient characteristics and their possibly different responses to Lokomat therapy could potentially have influenced the results.

Secondly, only chronic patients were included in the study. Therefore, no big changes between the assessments of walking ability were seen, which made it difficult to observe changes in robotic support during the course of the study.

Another limitation was the fact that AGS training sessions were not controlled for BWS. The therapist tried to maintain the same level of BWS throughout the study period, but it could not be guaranteed that the same levels were applied in all sessions. This may have influenced the robotic support during AGS training which could be a reason for the high variability in the AGS data.

Lastly, the AGS controller was only applied to one leg at a time, potentially leading to compensatory strategies to cope with a challenge on a single leg. While adapting the robotic support for both legs at the same time might prove too challenging for some patients, future analyses should address the effect of this choice.

## 5. Conclusions

This study investigated whether the long-term application of AGS, which automatically adapts the robotic support of the Lokomat, is feasible in patients with diverse neurological diseases. AGS was shown to be feasible in practice with non-ambulatory and ambulatory patients as a training tool and as an assessment tool. As a result of the targeted support provided to different gait phases, on average, AGS reduced the robotic support during Lokomat training, while giving feedback on the screen on which phase(s) required more support. It has been shown that when AGS was used as an assessment tool, a relationship between the data provided by AGS and conventional walking tests could be shown. AGS could be used to assess both non-ambulatory and ambulatory patients, offering the possibility of integrating the assessment in the training procedure and providing an objective and safe measurement. Nevertheless, this study provides further insights into the application of AGS as an assessment tool and indicates the need for further research in this area, particularly including large sample sizes in a multicenter setting. Additionally, a prospective study with non-ambulatory patients might reveal whether AGS is able to assess how far a patient is from being able to walk again.

## Figures and Tables

**Figure 1 brainsci-13-00612-f001:**
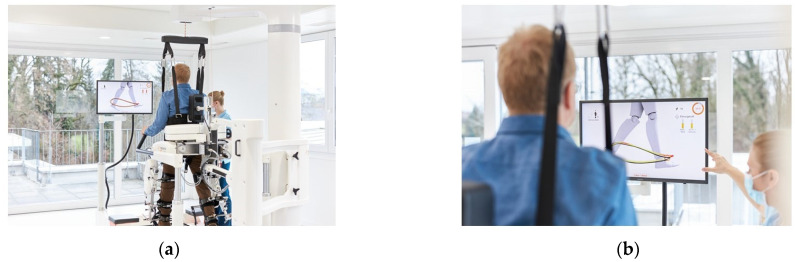
(**a**) Patient training with adaptive gait support (AGS) during Lokomat training. (**b**) AGS display. Image credit: Hocoma AG, www.hocoma.com.

**Figure 2 brainsci-13-00612-f002:**
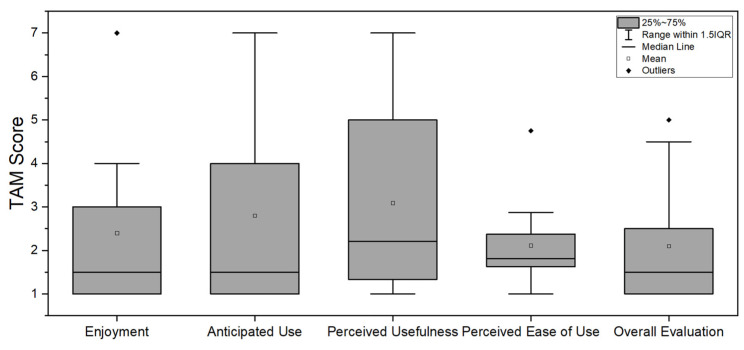
Summary of results of all TAM categories. The boxplot shows the mean answer for every category. Patients could react to the statements on a scale from 1 to 7. A lower score was associated with a good acceptance of AGS. The mean score for all categories was between the neutral and the positive associated answer. AGS: adaptive gait support; TAM: technology acceptance model.

**Figure 3 brainsci-13-00612-f003:**
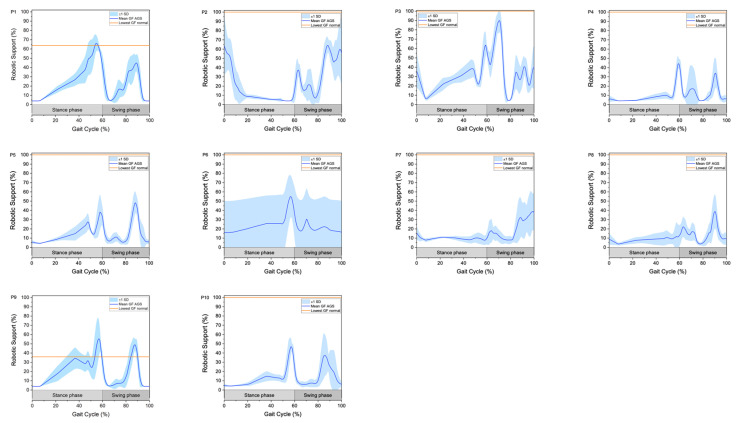
Robotic support (normalized stiffness averaged among hip and knee) over the gait cycle for all patients during one session of AGS training. The blue line indicates the mean robotic support of the 100th step of one leg (the better leg of the patient), for every training session of this patient. The 100th step was chosen because for every patient it was performed after the plateau in robotic support was reached. Light blue shows ±1 standard deviation (SD) over the total amount of gait cycles which were conducted for 4 min. The orange line represents the lowest robotic support value the therapist set for this patient throughout the study period.

**Figure 4 brainsci-13-00612-f004:**
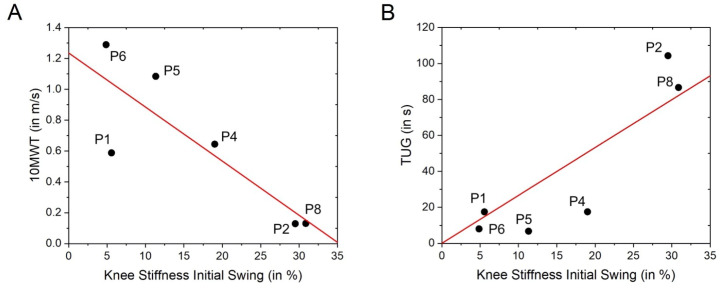
Results of the analysis of the best variable to explain walking ability with aggregated data over all ambulatory patients (each point indicates the mean of the three assessments for every patient). The red line indicates the regression line. The relationships between (**A**) 10MWT [m/s] and the variable knee stiffness initial swing (b = −0.04, *p*-value = 0.03, R^2^ = 0.71) and (**B**) TUG [s] and knee stiffness initial swing (b = 3.38, *p*-value = 0.02, R^2^ = 0.80) are represented. MWT: meter-walk test; TUG: timed up and go test.

**Figure 5 brainsci-13-00612-f005:**
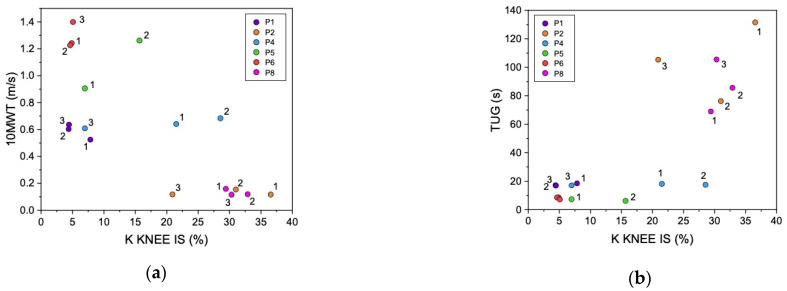
Knee stiffness at initial swing (K KNEE IS (%)) versus clinical assessments 10MWT (**a**) and TUG (**b**). Three data points per patient are shown (1st, 2nd, and 3rd assessment) in the same color.

**Figure 6 brainsci-13-00612-f006:**
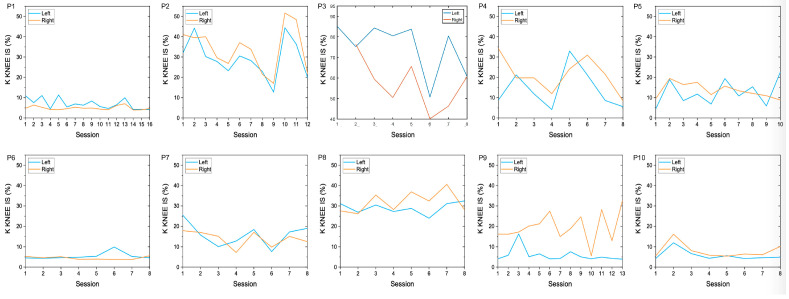
The knee stiffness of the initial swing phase (K KNEE IS (%)), which was chosen as the best variable to predict walking ability, for every training session of every patient. The value of the right leg of the first session of patient three was excluded. This patient experienced several error stops in the first training session with this leg and therefore it was not feasible to obtain a meaningful variable.

**Figure 7 brainsci-13-00612-f007:**
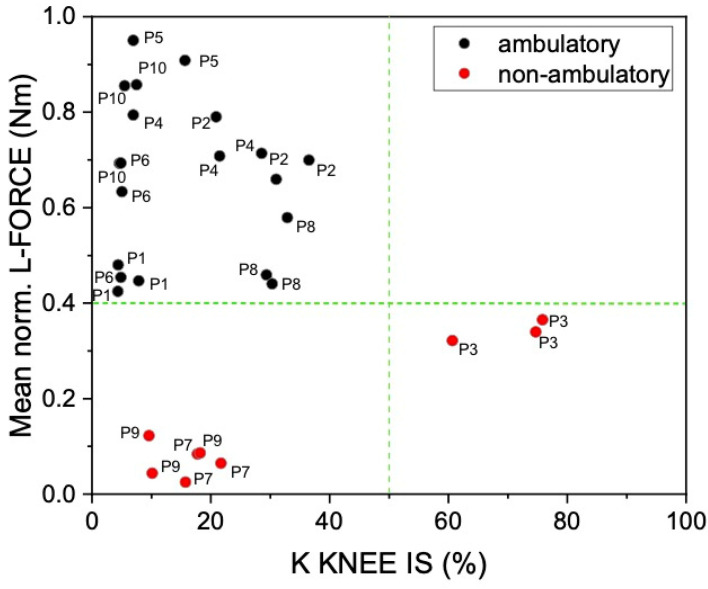
The knee stiffness at initial swing (K KNEE IS (%)) plotted against the mean normalized L-FORCE. For each patient, the data points of all three assessment sessions are shown. No linear relationship can be seen between the mean normalized L-FORCE measures and AGS data. Three clusters can be identified: non-ambulatory patients (red), despite all L-FORCE values being < 0.4 Nm/kg, can present with a high or low value of K KNEE IS (%). Ambulatory patients (black) show low values of K KNEE IS (%) and high values of L-FORCE.

**Table 1 brainsci-13-00612-t001:** Demographic and clinical features of patients.

PID	Diagnosis	Time Since Diagnosis	Gender (m/f)	Age (Years)	BMI (Kg/m^2^)	FAC (Level)
1	Aorta dissection	10 months	m	44	24.38	1
2	CLIPPERS	7 years	m	45	20.57	1
3	Epidural abscesses	18 months	f	40	25.54	0
4	Stroke	6 years	f	71	28.40	4
5	Viral infection	19 years	m	20	18.21	4
6	Cerebral palsy	26 years	f	26	20.50	5
7	Multiple sclerosis	20 years	f	58	15.63	0
8	Stroke	22 months	m	79	30.06	1
9	Neurosarcoidosis	7 years	m	65	26.09	0
10	Friedreich’s ataxia	13 years	f	53	21.51	2

BMI: body mass index; CLIPPERS: chronic lymphocytic inflammation with pontine perivascular enhancement responsive to steroids syndrome; FAC: Functional Ambulation Category; PID: patient identification number.

**Table 2 brainsci-13-00612-t002:** Number of conducted training sessions per participant.

PID	1	2	3	4	5	6	7	8	9	10
Num. of training sessions	16	12	8	8	10	8	8	8	13	8

PID: patient identification number.

**Table 3 brainsci-13-00612-t003:** Summary of conducted tests in the three assessment sessions.

	Assessment 1					Assessment 2				Assessment 3				
PID	MoCA	10MWT	TUG	L-FORCE	Quality of Life Questionnaires	10MWT	TUG	L-FORCE	Quality of Life Questionnaires	10MWT	TUG	L-FORCE	Quality of Life Questionnaires	TAM
1	x	x	x	x	x	x	x	x	x	x	x	x	x	x
2	x	x	x	x	x	x	x	x	x	x	x	x	x	x
3	x			x	x			x	x			x	x	x
4	x	x	x	x	x	x	x	x	x	x	x	x	x	x
5	x	x	x	x	x					x	x	x	x	x
6	x	x	x	x	x	x	x	x	x	x	x	x	x	x
7	x			x	x			x	x			x	x	x
8	x	x	x	x	x	x	x	x	x	x	x	x	x	x
9	x			x	x			x	x			x	x	x
10	x			x	x			x	x			x	x	x

MoCA: Montreal Cognitive Assessment; MWT: meter-walk test; TAM: technology acceptance model; TUG: timed up and go test; PID: patient identification number; x: test was conducted. Subject 10, despite being able to walk, did not conduct the walking assessments for issues related to therapy organization.

**Table 4 brainsci-13-00612-t004:** *p*-values of the regression model analysis for 10MWT and each parameter.

**10MWT~1**	0.0214											
	**Stiffness**						**Damping**					
	**Knee**			**Hip**			**Knee**			**Hip**		
	**Initial Swing**	**Mid Swing**	**Terminal Swing**	**Initial Swing**	**Mid Swing**	**Terminal Swing**	**Initial Swing**	**Mid Swing**	**Terminal Swing**	**Initial Swing**	**Mid Swing**	**Terminal Swing**
**10MW~variable**	0.03429	0.907	0.1439	0.459	0.4535	0.1258	0.9441	0.877	0.844	0.388	0.9154	0.9397

**Table 5 brainsci-13-00612-t005:** *p*-values of the regression model analysis for TUG and each parameter.

**TUG~1**	0.0733											
	**Stiffness**						**Damping**					
	**Knee**			**Hip**			**Knee**			**Hip**		
	**Initial Swing**	**Mid Swing**	**Terminal Swing**	**Initial Swing**	**Mid Swing**	**Terminal Swing**	**Initial Swing**	**Mid Swing**	**Terminal Swing**	**Initial Swing**	**Mid Swing**	**Terminal Swing**
**TUG~variable**	0.016	0.680	0.0407	0.674	0.596	0.0353	0.694	0.782	0.444	0.3865	0.658	0.541

**Table 6 brainsci-13-00612-t006:** Average in percentage change in normalized knee stiffness in the initial swing phase (K KNEE IS) of the left and right leg from the first to last session.

PID	1	2	3	4	5	6	7	8	9	10
change in normalized K KNEE IS (%)	−3.4	−15.6	−15.2	−14.5	+8.7	+0.2	−5.9	+0.9	+8.1	+2.6

## Data Availability

The datasets used and/or analyzed during the current study are available from the corresponding author upon reasonable request.

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
