# Peer review of "Feasibility of an Intelligent Algorithm Based on an Assist-as-Needed Controller for a Robot-Aided Gait Trainer (Lokomat) in Neurological Disorders: A Longitudinal Pilot Study"

_brainsci, 2023, doi:10.3390/brainsci13040612_

Round 1
Reviewer 1 Report
Comments and Suggestions for Authors The article presents a study evaluating adaptive support to overcome the constant assistanceprovided by traditional robotic walking devices. Specifically, the Adaptive Gait Support (AGS)
controller for the Lokomat exoskeleton was investigated in a longitudinal study involving a
heterogeneous group of neurological subjects with walking disabilities. This study had several
objectives: to test the feasibility of AGS in a long-term application, to highlight the differences
between adaptive support and manually set support, and to investigate the relationship between
adaptive support and traditional walking tests.
The work is interesting and addresses an issue relevant to personalize treatments according to
patient’s abilities. However, to improve the quality and impact of their study, the authors should
address the following points.
1) The main criticism concerns the number of subjects involved. Ten subjects are few to obtain
significant results (as stated in the Limitations section by the authors themselves). All the more so
if 4 of the ten subjects did not comply with the experimental protocol (training interruption,
dropouts, inability to perform over-ground walking) as the others did. Why was the study not
extended to other subjects to standardize the experiment? This extension would have provided
more data and improved the results. For example, the results on the relationship with the
10MWT and TUG tests (Figure 4) rely on only six subjects. The same is valid for Figure 5 and
probably for the results of the TAM questionnaire (Figure 2), as not all participants complete the
experimental protocol. The first suggestion for the authors is to increase the number of subjects
involved in the study or at least that all participants comply with and complete the experimental
protocol.
2) The Introduction is well-written. However, more studies and reviews on using Lokomat are
available in the literature. The authors should expand the Introduction by adding more up-to-
date references on the Lokomat and its applications in clinical trials.
3) Participants (Paragraph 2.2): Table 1 shows that the ten patients have very different diagnoses
and clinical conditions. They all have walking problems, but their responses to rehabilitation
and/or assistive treatments could differ. Could this choice have limited the final results,
representing a weakness rather than a strength of the study? Did it not introduce bias into the
results?
4) Line 108: The authors should better explain the sentence “The gait cycle was divided into 30
windows w.”
5) Line 110: The authors should provide more detail on “The error metric allows deviations from the
reference trajectory within predefined dead-bands”. Were these predefined limits set equally for
all participants?
6) Line 119: Has the minimum safety threshold (4%) been set equally for all participants?
7) Lines 121-122: Could adaptive support on one leg and maximum constant support (100%) on the
other have caused participants to perceive a sense of "imbalance" during AGS training?
8) Lines 137-139: The authors should clarify the sentences on BWS.
9) Lines 161-164: The authors should expand the section on data analysis because it needs to
provide the necessary information on data processing. In addition to stiffness and damping data,
does the Lokomat provide data/measures/signals on gait cycles? How were they analyzed with
Matlab to determine the gait phases? More details on this point should be provided in a
dedicated subsection.
10) Lines 166-167: More details on the structure/items of the TAM questionnaire and the scoring
procedure should be included.
11) Line 170: Why only 4 minutes? An explanation of this choice should be added.
12) Line 191: In the previous step (lines 187-188), 12 values were available for each leg (12 for the
left and 12 for the right). Then, the authors averaged them. The procedure needs to be better
explained to clarify how the authors obtained six final values for stiffness and 6 for damping.
13) Lines 194-196: Some participants probably had different levels of impairment severity in the two
legs. Why did the authors average the L-FORCE values for the two legs? Also, why did the authors
average the different movements? An ad-hoc analysis of leg impairment and specific movement
would have been more significant.
14) Section 2.6 – Statistical Analysis: The authors should expand all subsections to include details and
rationale on statistical tests, variables involved, and methodological approach.
15) Section 3.1 - Participants: Participants ranged from underweight to obesity conditions (based on
BMI ranges) and from the inability to walk on the ground to independent walking: how might this
variety of physical conditions, regardless of pathology, have influenced the results obtained? The
authors should provide explanations on this point. Also, Table 1 should indicate patients unable
to perform over-ground walking (but why have they been included if the study aims to test the
correlation of AGS with 10MWT and TUG tests?). It would also be helpful to add a summary table
(at the end of section 3.1) that highlights which tests were administered to each patient (for each
assessment session): since some participants were unable to perform the walking tests and
others discontinued the protocol, this table would be necessary to understand the subsequent
analysis sections better. Information on the number of training sessions performed during the
eight weeks should also be reported.
16) Section 3.2 - Feasibility: This section should be expanded with more detail. In line 260, the
authors state that “the mean score for every question” is shown in Figure 2, but the TAM
questionnaire traditionally includes more than five questions. The caption in Figure 2 states
“category”: the authors should indicate which TAM items were considered for each category
shown in Figure 2. In line 261, the authors indicate that “8/10 patients would use AGS again”: was
the TAM questionnaire administered to all participants (e.g., it appears that P5 did not complete
the protocol)? The authors could also discuss why two subjects would not use AGS again (too
challenging?).
17) Figure 3: The quality of the figures should be improved. The axis “Robotic Support (%)” should be
the same in all figures (0-100) for a fair comparison. Why was the 100th step chosen as
representative of the training session? Why was the best leg chosen and not the most impaired
leg? Figure P6 shows the broader standard deviation in Robotic Support: this seems incongruent
considering that P6 has the best FAC score (FAC=5). The same is true for the support set manually
by the therapist (close to 100%), which is higher than that of P9 (less than 40%) with FAC=0. In
addition, the percentage of stance and swing phases is the same for all patients: is this a
constraint imposed by the Lokomat? The percentages of stance and swing phases typically differ
depending on the severity of gait impairment. The authors should discuss these points.
18) Lines 285-287: On what basis, among the 12 estimated parameters, did the authors choose to use
only “knee stiffness initial swing phase” for the analysis? The authors should justify this choice.
Also, it is unclear why estimate 12 parameters if only one was considered for the analysis.
19) Figure 4: Adding the PID to each point in the figure would be helpful. In addition, the caption
should describe the figure and not discuss the results (lines 292-301). These lines should be
moved after the figure. Finally, it would be appropriate to show the regression model data for
each parameter and walking test in a table to justify the choice of “knee stiffness initial swing
phase” as the best variable (see comment 18). Why did the authors exclude the damping
variables? For completeness, they could/should be included in the new table.
20) Figure 5: Including a legend associating the colors and PID would be helpful. Also, the caption
should describe the figure rather than discuss the results (lines 307-310). These lines should be
moved after the figure.
21) Figure 6: As in Figure 3, PID should be added next to each graph. For fair comparison, the “K
KNEE IS (%)” axis should be the same in all figures. In P3 (first row, third graph), the first point for
the right leg seems missing. Also, the caption should describe the figure and not discuss the
results (lines 317-327). These lines should be moved after the figure. Finally, it would be
interesting to report the percentage of improvement (worsening) between the first and last AGS
session for each participant.
22) Figure 7: The distinction between ambulatory and non-ambulatory patients emerges here for the
first time in the paper. The authors should explain these two categories. In addition, it would be
helpful to add the PID to each point shown in the figure.
23) Lines 345-348: The methodological approach for reliability should be clarified in section 2.6.5,
providing more details and explanations: this allows an easy understanding of the statistical
significance of results in section 3.3.5, which, however, should be expanded.
24) The Discussion and Conclusions sections should be revised according to the required changes and
should be more focused on the study results.
25) The Limitations section seems to suggest that a more constrained protocol and a larger number
of patients would allow for more meaningful results. The authors should consider revising the
experimental protoco

Author Response
Zurich, 24th March 2023
Mrs. Della Diao
Dear Editor,
I wish to thank you for handling our manuscript entitled “Feasibility of an intelligent algorithm based on an Assist-As-Needed Controller for a Robot-Aided Gait Trainer (Lokomat) in Neurological Disorders: a longitudinal pilot-study” as well as we thank the reviewers for their constructive feedback.
Please find enclosed our detailed replies to the comments of the reviewers.
We hope you and the reviewers will find our revision satisfactory and the manuscript can now be accepted for publication on Brain Sciences.
Thank you again for your kind attention and assistance.
With best regards
Daniele Munari
RESPONSE TO REVIEWERS
We would like to express our appreciation for the time and effort put in by the editor and the reviewers. Their comments and suggestions have improved the quality of the manuscript.
Response to Reviewer #1
Comments to the Author
Thank you for allowing me to review this manuscript. You will find my comments below.
- Response to Reviewer #1 comment: - The main criticism concerns the number of subjects involved. Ten subjects are few to obtain significant results (as stated in the Limitations section by the authors themselves). All the more so if 4 of the ten subjects did not comply with the experimental protocol (training interruption, dropouts, inability to perform over-ground walking) as the others did. Why was the study not extended to other subjects to standardize the experiment? This extension would have provided more data and improved the results. For example, the results on the relationship with the 10MWT and TUG tests (Figure 4) rely on only six subjects. The same is valid for Figure 5 and probably for the results of the TAM questionnaire (Figure 2), as not all participants complete the experimental protocol. The first suggestion for the authors is to increase the number of subjects involved in the study or at least that all participants comply with and complete the experimental protocol.”
- Authors’ reply: We understand the concerns of the Author, and we would like to clarify this point. The main aim of the project was to test the new algorithm (implemented in the Lokomat commercial version in 2020) by planning and conducting a feasibility study. Differently from the study of Maggioni [1], which was conducted with a research version of the software, the current study was the first one to test the commercially available software in clinical practice. While trying to standardize the protocol, we took the approach of a real-world evidence study, including all patients training with the Lokomat in the rehabilitation center who met our relatively broad inclusion criteria. This is reason why we enrolled subjects suffering from different neurological diseases. We decided to include also non-ambulatory subjects (despite not being able to obtain 10MWT and TUG results from them), as the main target group of the Lokomat consists precisely of non-ambulatory patients and we were very interested in understanding the feasibility of the adaptive algorithm in this population. We are aware that this poses a sort of conundrum, as here we cannot validate the AAN results against any walking test, but we hoped to get meaningful insights from the isometric force assessment at least (see Figure 7). As mentioned in paper, the small sample size is one of the limitations of this study and in the conclusion section we provided future directions and some insights for a future multicenter study.
- Response to Reviewer #1 comment - The Introduction is well-written. However, more studies and reviews on using Lokomat are available in the literature. The authors should expand the Introduction by adding more up-to-date references on the Lokomat and its applications in clinical trials.
- Authors’ reply: According to the suggestion of Reviewer #1, we added the following more up-to-date references on the Lokomat and its applications in clinical trials: “Baronchelli et al., The Effect of Robotic Assisted Gait Training With Lokomat® on Balance Control After Stroke: Systematic Review and Meta-Analysis. Front Neurol. 2021, Zhang et al., C. Comparison of Efficacy of Lokomat and Wearable Exoskeleton-Assisted Gait Training in People With Spinal Cord Injury: A Systematic Review and Network Meta-Analysis. Front Neurol. 2022 and Calabrò et al., Italian Consensus Conference on Robotics in Neurorehabilitation (CICERONE). What does evidence tell us about the use of gait robotic devices in patients with multiple sclerosis? A comprehensive systematic review on functional outcomes and clinical recommendations. Eur J Phys Rehabil Med. 2021
- Response to Reviewer #1 comment - Participants (Paragraph 2.2): Table 1 shows that the ten patients have very different diagnoses and clinical conditions. They all have walking problems, but their responses to rehabilitation and/or assistive treatments could differ. Could this choice have limited the final results, representing a weakness rather than a strength of the study? Did it not introduce bias into the results?.
- Authors’ reply: We would like to thank the Author for the comment. Since this is a feasibility study, we decided to include patients with different neurological diseases for understanding the applicability of the novel Lokomat application in a wide range of clinical conditions, which is what clinicians in a standard clinical setting are normally confronted with. This was also due to pragmatic reasons, as the outpatient clinic where the study took place treated a heterogenous patient population. We agree that patients’ response to treatment could differ based on the pathology and on clinical symptoms such as spasticity, reduced sensory feedback etc., and sub-group analyses should be carried out in future studies. We think that the results of the study were certainly affected by the wide variety of clinical symptoms, but that this rather added noise to the data than bias. We do think that having a more homogeneous cohort would have resulted in more definite results. We had mentioned in the Limitations that with the current number of patients, a sub-group analysis was not possible. In Future directions, we added that large-scale studies that allow sub-group analyses could provide more insights on the recovery trajectories of specific diseases.
- Response to Reviewer #1 comment - Line 108: The authors should better explain the sentence “The gait cycle was divided into 30 windows w.”
- Authors’ reply: According to the suggestion of Reviewer #1 we changed the sentence as follow “To optimally support each gait phase, 30 windows w per step are implemented and the impedance is adapted separately in each window [15].”
- Response to Reviewer #1 comment - Line 110: The authors should provide more detail on “The error metric allows deviations from the reference trajectory within predefined dead-bands”. Were these predefined limits set equally for all participants?
- Authors’ reply: The dead-bands were set equally for all participants. According to the suggestion of Reviewer #1, we modify the sentence as follows: “The error metric allows deviations from the reference trajectory within predefined deadbands around the hip and knee reference trajectories, that were set equally for all participants.” We also refer to another paper from one of the co-authors for details on the technical implementation of the algorithm [1].
- Response to Reviewer #1 comment - Line 119: Has the minimum safety threshold (4%) been set equally for all participants?
- Authors’ reply: yes, the minimum safety threshold (4%) was set equally for all participants by the therapist.
- Response to Reviewer #1 comment - - Lines 121-122: Could adaptive support on one leg and maximum constant support (100%) on the other have caused participants to perceive a sense of "imbalance" during AGS training?
- Authors’ reply: We think this is a valid point, but after some pilot tests we had chosen to adapt the robotic support on one leg at a time for two reasons: 1) safety: the most impaired patients could not cope with a simultaneous adaptation of the support on both legs. This caused the Lokomat safety mechanisms to be triggered too often resulting in frequent stops of the training. 2) it was not possible to provide visual feedback on the reference and actual trajectories for both legs at the same time. The screen only shows feedback on one leg at a time (see Fig. 1b).
Participants were used to train with Lokomat device and before participating in the study, all subjects completed two familiarization sessions with the new algorithm, which also allowed them to experience these different settings for left and right leg. No participant reported any sense of imbalance or discomfort in the lower limbs.
- Response to Reviewer #1 comment - - Lines 137-139: The authors should clarify the sentences on BWS.
- Authors’ reply: According to the suggestion of Reviewer #1, we added the following clarification: “No fixed value for the body weight support (BWS) provided by the Lokomat was defined, although the therapist was instructed to ensure that the patient trained in the dynamic range of the BWS system (i.e. where the BWS system ensures a constant unloading throughout the gait cycle). The amount of BWS was set to guarantee an upright body position and to avoid knee buckling during stance, ensuring at the same time a challenging training in a safe environment. The therapist was also asked to avoid large ses-sion-to-session changes in BWS.”
- Response to Reviewer #1 comment - Lines 161-164: The authors should expand the section on data analysis because it needs to provide the necessary information on data processing. In addition to stiffness and damping data, does the Lokomat provide data/measures/signals on gait cycles? How were they analyzed with Matlab to determine the gait phases? More details on this point should be provided in a dedicated subsection.)”.
Authors’ reply: The Lokomat provides also the reference and actual joint trajectories for hip and knee. According to the suggestion of Reviewer #1, we added more information on the data processing in section 2.5 Data analysis. In particular, the gait cycle was divided in phases according to recommendations found in gait literature, applied to the Lokomat reference hip and knee trajectories [2, 3]. We are aware that this is an approximation but given that participants walked in the Lokomat following a trajectory imposed by the device with a fixed timing, we believe that dividing the gait cycle according to predefined phases (initial swing: 10.5% gait cycle, mid swing: 14% gait cycle and terminal swing: 13% gait cycle) is acceptable for our context. The points of transition from swing to stance were suggested by the Lokomat user information.
- Response to Reviewer #1 comment: Lines 166-167: More details on the structure/items of the TAM questionnaire and the scoring procedure should be included.
- Authors’ reply: According to the suggestion of Reviewer #1, we added more details regarding the structure/items of the TAM questionnaire and the scoring procedure as follow “Five determinants were identified as related to technology acceptance. The determinants are (a) perceived usefulness, (b) perceived ease of use, (c) perceived enjoyment, (d) intention of use and (e) self-efficacy. Likert scales from 1 to 7 (1 is strongly disagree and 7 is strongly agree) were used.
- Response to Reviewer #1 comment - Line 170: Why only 4 minutes? An explanation of this choice should be added.
- Authors’ reply: The first minute was excluded from the calculation as the normalized stiffness adapted according to Eq. 1 required some time to decrease (exponential decay) until a point where the patient was actually able to influence the gait trajectory (i.e. when the robotic stiffness is too high, the patient’s hip and knee are forced to follow the reference trajectory). According to the comments of Reviewer #1, we modify the sentence as follow “The first minute was excluded from the calculation as the normalized stiffness adapted according to Eq. 1 required some time to decrease (exponential decay) until a point where the patient was actually able to influence the gait trajectory.”
- Response to Reviewer #1 comment - Line 191: In the previous step (lines 187-188), 12 values were available for each leg (12 for the left and 12 for the right). Then, the authors averaged them. The procedure needs to be better explained to clarify how the authors obtained six final values for stiffness and 6 for damping.
- Authors’ reply: According to this comment of Reviewer #1, we added and explanation and modified the section as follow: “This approach resulted in 12 different values of stiffness/damping for each leg and per training session. For the comparison of the AGS variables with the gait assessments, the calculated values were averaged over the left and right leg for every individual assessment session, which resulted in 6 final stiffness (knee stiffness of all 3 gait phases and hip stiffness of all 3 gait phases averaged over left and right leg) and 6 final damping values (knee damping of all 3 gait phases and hip damping of all 3 gait phases averaged over left and right leg).”
- Response to Reviewer #1 comment - Lines 194-196: Some participants probably had different levels of impairment severity in the two legs. Why did the authors average the L-FORCE values for the two legs? Also, why did the authors average the different movements? An ad-hoc analysis of leg impairment and specific movement would have been more significant...
- Authors’ reply: We thank the Reviewer #1 for the comment. We agree that it is better to explore the relationship between AGS data and L-FORCE separately for the left and right leg. We redid the analysis of the L-FORCE correlation with AGS data for both legs separately. For every leg individually we averaged the L-FORCE as well as the AGS data of every patient for all three assessment sessions. The L-FORCE data of the left leg showed no significant relationship with the AGS variables of the left leg (p>0.05). The analysis of the data of the right leg resulted in the same outcome: no relationship was found between AGS data and L-FORCE data (p>0.05).
We decided instead to present the results of the average L-FORCE across movements (flexion/extension, hip/knee), after several attempts of correlating specific movements/direction with the AGS data from the relative phases of the gait cycle when a movement is performed led to no significant results.
- Response to Reviewer #1 comment - Section 2.6 – Statistical Analysis: The authors should expand all subsections to include details and rationale on statistical tests, variables involved, and methodological appr.
- Authors’ reply: According to this comment of Reviewer #1 we expanded the section 2.6 Statistical analysis, by adding more information in the different sub-sections. We removed the analysis on the within-patient effect (Non-aggregated data) as the low number of subjects limited the applicability of the mixed effect model approach.
- Response to Reviewer #1 comment - Section 3.1 - Participants: Participants ranged from underweight to obesity conditions (based on BMI ranges) and from the inability to walk on the ground to independent walking: how might this variety of physical conditions, regardless of pathology, have influenced the results obtained? The authors should provide explanations on this point. Also, Table 1 should indicate patients unable to perform over-ground walking (but why have they been included if the study aims to test the correlation of AGS with 10MWT and TUG tests?). It would also be helpful to add a summary table (at the end of section 3.1) that highlights which tests were administered to each patient (for each assessment session): since some participants were unable to perform the walking tests and others discontinued the protocol, this table would be necessary to understand the subsequent analysis sections better. Information on the number of training sessions performed during the eight weeks should also be reported.
- Authors’ reply: We would like to thank the Author for the comment. Since this a feasibility study conducted in an outpatient facility treating various conditions, we decided to include a broad range of pathologies to understand if the novel software was feasible for all (see also answer to comment 1). In Table 1, FAC = 0 indicates inability to walk. These subjects were included anyway, even if not able to perform the walking tests, as it was very important for us to understand if the novel AGS software was feasible for the most severe patients, who are the main target population of the Lokomat. According to this comment of Reviewer #1, we expanded this point in the limitation section. Then, we included tables where we reported the number of training sessions performed during the eight weeks. Lastly, the administered tests were also added in chapter 3.1.
- Response to Reviewer #1 comment - Section 3.2 - Feasibility: This section should be expanded with more detail. In line 260, the authors state that “the mean score for every question” is shown in Figure 2, but the TAM questionnaire traditionally includes more than five questions. The caption in Figure 2 states “category”: the authors should indicate which TAM items were considered for each category shown in Figure 2. In line 261, the authors indicate that “8/10 patients would use AGS again”: was the TAM questionnaire administered to all participants (e.g., it appears that P5 did not complete the protocol)? The authors could also discuss why two subjects would not use AGS again (too challenging?)
- Authors’ reply: we would like to thank of Reviewer #1 for his/her comments and we would like to explain that the TAM questionnaire was administered to all participants. P5 did not complete the protocol, he wanted to withdraw from the study because of COVID-19. However, we were able to do a closing session with the patient where all walking tests, the L-FORCE and the TAM were conducted. This patient still has more than 8 training sessions with AGS, only 1 assessment session of walking tests and L-FORCE is missing. We only used 5 TAM items because as guideline we used the papers from F. D. Davis (F. D. Davis, “A Technological Acceptance Model for empirically testing new end-user information systems: theory and results,” Ph.D. dissertation, 1985, MIT Sloan School of Management) and (V. Venkatesh and F. D. Davis, “A Theoretical Extension of the Technology Acceptance Model: Four Longitudinal Field Studies,” Management Science, vol. 46, no. 2, pp. 186–204, 2000) where he invented 4 major items for the TAM questionnaire.
- Response to Reviewer #1 comment - Figure 3: The quality of the figures should be improved. The axis “Robotic Support (%)” should be the same in all figures (0-100) for a fair comparison. Why was the 100th step chosen as representative of the training session? Why was the best leg chosen and not the most impaired leg? Figure P6 shows the broader standard deviation in Robotic Support: this seems incongruent considering that P6 has the best FAC score (FAC=5). The same is true for the support set manually by the therapist (close to 100%), which is higher than that of P9 (less than 40%) with FAC=0. In addition, the percentage of stance and swing phases is the same for all patients: is this a constraint imposed by the Lokomat? The percentages of stance and swing phases typically differ depending on the severity of gait impairment. The authors should discuss these points...
- Authors’ reply: according to this comment, we increased the quality of Fig. 3 and added the following sentence in the caption “The 100th step was chosen because for every patient it was performed after the plateau in robotic support was reached.”
The percentage of stance and swing phases typically differ depending on the severity of the gait impairment, when walking overground. However, the Lokomat drives the patient’s leg along a predefined gait trajectory, imposing a certain percentage of swing and stance phase. We divided the gait cycle and swing and stance according to what is report in the Lokomat user information.
- Response to Reviewer #1 comment - Lines 285-287: On what basis, among the 12 estimated parameters, did the authors choose to use only “knee stiffness initial swing phase” for the analysis? The authors should justify this choice. Also, it is unclear why estimate 12 parameters if only one was considered for the analysis..
- Authors’ reply: According to this comment of Reviewer #1 we expanded the explanation on the statistics in section 2.6.2 (forward variable selection model and Elastic Net regression) and also the explanation of the results in section 3.3.2 as follow: “In both models, the variable knee stiffness initial swing phase was chosen with the forward selection method as the best variable to explain the variation in walking ability, (see Figure 4). Only one variable was chosen because it was the only variable which whose P-value was <.05. Therefore, no other additional variable was added to the model. The variable knee stiffness initial swing phase was also the most frequently selected by the Elastic Net method. ”
We included 12 variables in the analysis because we did not know a-priori which one(s) were the most useful to explain gait ability as measured with the walking tests. Therefore, we used a variable selection method (forward selection) and, according to the suggestions of reviewer #1, we recently confirmed this result using the Elastic Net method.
- Response to Reviewer #1 comment - Figure 4: Adding the PID to each point in the figure would be helpful. In addition, the caption should describe the figure and not discuss the results (lines 292-301). These lines should be moved after the figure. Finally, it would be appropriate to show the regression model data for each parameter and walking test in a table to justify the choice of “knee stiffness initial swing phase” as the best variable (see comment 18). Why did the authors exclude the damping variables? For completeness, they could/should be included in the new table
- Authors’ reply: According to this comment of Reviewer #1 we modified figure 4 and moved the discussion of the results after the figure. We integrated the regression model p-value for each parameter and the two walking tests in tables 4 and 5. We added the sentence to chapter 3.3.2 “Only one variable was chosen because in step 2 of the stepwise variable selection, no additional variable could be selected for the model (Tables 4 and 5). The variable knee stiffness initial swing phase was also the most frequently selected by the Elastic Net method. No damping variable was selected with the forward selection method nor with the Elastic Net.
- Response to Reviewer #1 comment - Figure 5: Including a legend associating the colors and PID would be helpful. Also, the caption should describe the figure rather than discuss the results (lines 307-310). These lines should be moved after the figure...
- Authors’ reply: According to this comment of Reviewer #1, we added a legend in Figure 5 and moved the discussion of the figure after the figure. Thank you for pointing this out.
- Response to Reviewer #1 comment - Figure 6: As in Figure 3, PID should be added next to each graph. For fair comparison, the “K KNEE IS (%)” axis should be the same in all figures. In P3 (first row, third graph), the first point for the right leg seems missing. Also, the caption should describe the figure and not discuss the results (lines 317-327). These lines should be moved after the figure. Finally, it would be interesting to report the percentage of improvement (worsening) between the first and last AGS session for each participant...
- Authors’ reply: According to this comment of Reviewer #1 we added a sentence in the figure description: “The value of the right leg of the first session of patient 3 was excluded. This patient experienced several error stops in the first training session with this leg and therefore it was not feasible to obtain a meaningful variable.” Furthermore, PIDs were added in the figures and table 6 was introduced to show the change in knee stiffness of the initial swing phase from the first to the last AGS session for each patient.
- Response to Reviewer #1 comment - Figure 7: The distinction between ambulatory and non-ambulatory patients emerges here for the first time in the paper. The authors should explain these two categories. In addition, it would be helpful to add the PID to each point shown in the figure...
- Authors’ reply: According to this comment of Reviewer #1, PIDs in the figure were added. Thank you for pointing out the missing information about the non-ambulatory subjects: we added a sentence in section 1.
- Response to Reviewer #1 comment - Lines 345-348: The methodological approach for reliability should be clarified in section 2.6.5, providing more details and explanations: this allows an easy understanding of the statistical significance of results in section 3.3.5, which, however, should be expanded.
- Authors’ reply: According to this comment of Reviewer #1, we expanded the description of the methodological approach in section 2.6.5: “Absolute reliability was assessed using the Bland-Altman plot with Limits of Agreements (LOAs). The Limits of Agreements (LOAs) indicate the range where, for a new person from the studied population, the difference between any two tests lies within a 95% probability. Changes between two measurements are considered significant only if they fall outside the LOAs [4].” We also added a further explanation of the results in 3.3.5. Additionally, we removed the intraclass coefficients (ICC) based on ANOVA in order to only present one statistical test which investigates relative reliability.
- Response to Reviewer #1 comment - The Discussion and Conclusions sections should be revised according to the required changes and should be more focused on the study results.
- Authors’ reply: According to comments and requests from the Reviewer #1 and the Reviewer #2, we revised the discussions and conclusions sections.
- Response to Reviewer #1 comment - The Limitations section seems to suggest that a more constrained protocol and a larger number of patients would allow for more meaningful results. The authors should consider revising the experimental protocol...
- Authors’ reply: We learned a lot from the current study and would introduce some changes in future studies. On one hand, the experimental protocol used in this study was flexible enough to be used in a real clinical setting and explore the feasibility of the novel AGS software in a broad range of patients (different diagnosis, stages and levels of disability). This could not have been possible with a more constrained protocol. On the other hand, drawing conclusions on psychometric properties of the assessment and on longitudinal progression of the patients was very challenging, leaving us with some hints on what to address next, but no conclusive results. We tried to address limitations, learning and ideas for future studies in the respective sections of the discussion.
- Maggioni, S., et al., Assessing Walking Ability Using a Robotic Gait Trainer: Opportunities and Limitations of Assist-as-Needed Control in Spinal Cord Injury. PREPRINT (Version 1) available at Research Square, 2022.
- Perry, J. and J.M. Burnfield, Gait Analysis: Normal and Pathological Function. Vol. 2. 2010, Thorofare, New Jersey: Slack Inc.
- Baker, R. GAIT GRAPHS FOR BEGINNERS. 2013 [cited 2016; Available from: https://wwrichard.net/tag/gait-graphs/.
- Atkinson, G. and A.M. Nevill, Statistical methods for assessing measurement error (reliability) in variables relevant to sports medicine. Sports medicine, 1998. 26: p. 217-238.
- Marchal-Crespo, L. and D.J. Reinkensmeyer, Review of control strategies for robotic movement training after neurologic injury. J Neuroeng Rehabil, 2009. 6: p. 20.
- Campagnini, S., et al., Effects of control strategies on gait in robot-assisted post-stroke lower limb rehabilitation: a systematic review. J Neuroeng Rehabil, 2022. 19(1): p. 52.
- Riener, R., et al., Patient-Cooperative Strategies for Robot-Aided Treadmill Training: First Experimental Results. IEEE Trans Neural Syst Rehabil Eng, 2005. 13(3): p. 380-94.
- Blank, A.A., et al., Current Trends in Robot-Assisted Upper-Limb Stroke Rehabilitation: Promoting Patient Engagement in Therapy. Curr Phys Med Rehabil Rep, 2014. 2(3): p. 184-195.
- de Miguel-Fernández, J., et al., Control strategies used in lower limb exoskeletons for gait rehabilitation after brain injury: a systematic review and analysis of clinical effectiveness. Journal of NeuroEngineering and Rehabilitation, 2023. 20(1): p. 23.
Reviewer 2 Report
Comments and Suggestions for Authors
The authors investigate the feasibility of assist-as-needed controlling strategies (with the Lokomat device) in long-term neurorehabilitation. Despite a small cohort, the main finding of the article is that Adaptive Gait Support provides less force to the patient during the gait, fostering autonomous walking. The finding is important also given the relationship between support levels and walking tests values.
MAJOR
General
- Was LokomatPro involved in the study design/results choice and publishing?
Introduction
- I have a small doubt on the aim of the work. The authors are investigating feasibility of a new algorithm in long-term rehabilitation, already demonstrated in two short-term sessions. However, a feasibility study is a study where an assessment of the practicality of a proposed project plan or method is tested. However, I do not clearly see the difference of testing the feasibility of an algorithm used ‘long’ or ‘short’ term. What I mean is the following:
1. The Lokomat has already been extensively validated for feasibility and also efficacy, both short- and long-term, thus the robot is not excessively problematic and it is usable in clinics.
2. The algorithm AGS feasibility has already been demonstrated in short term.
Now my question, does using an algorithm for a long time change the request that the user needs to have from the algorithm (i.e., why do we need to evaluate its feasibility again?).
I would suggest the author to better describe what is the difference from previous studies either on Lokomat and on AGS, better detailing which are the differences between short- and long- term rehabilitation that require a second assessment of feasibility.
Materials
- Methods and protocols are clearly explained and easy to follow.
- Line 204: Please, check whether data is normal, report the result, then apply the proper statistical test for paired data (t-test or Wilcoxon).
- Line 210: I understand that 12 variables and 10 subjects are not easy to handle in multivariate analysis. However, I would suggest to avoid any sort of multivariate variable selection with only 10 subjects. Such analysis, and in particular backward and forward regressions are not only sub-optimal, but often misleading. This is the case when two or more independent variables are strongly dependent and the collinearity hypothesis is not respected (most likely with 12 gait-related variables). However, I understand the author’s necessity to perform a multivariate ‘feature ranking’. I would suggest to use an optimized (ratio between L1 and L2 and regularization strength) Elastic-Net linear regression and use the coefficients/p-values of that regression, much more robust to collinearity.
- Line 213: Mixed effects linear models often require very high data cardinality. Did you perform an estimate of statistical power of these results? I understand the character of the analysis as exploratory and avoiding FDR correction. However, mixed effects introduce a third dimension in the analysis, thus requiring a much larger sample size. For what concern this analysis, I would suggest either removing or providing an estimate of effect size.
Results
- The participants are clearly described and I congratulate the authors for the patient-specific description of the enrollment phase.
- Line 289-290: From here I understand that the forward analysis has been performed on a dataset with some variables present for 10 patients while some other only for 6. This, often results in a list-wise elimination of subjects i.e., if one patient does not have a value for one variable, the whole patient is removed from the analysis, thus decreasing the cohort from 10 to 6 patients. If this is correct, the point raised at line 210 becomes even more crucial. With 6 patients I would avoid in any case multivariate analysis, and would only report univariate linear regressions between instrumental variables and outcomes.
- Figure 7 is a fundamental result of the analysis, I would give the emphasis of the work on this concept, more than on the feasibility of the algorithm.
MINOR
Introduction:
- Even if clear to me, I may suggest to provide around line 40-45 a quick introduction on different control strategies (Force, Position, etc…) in lower-limb robotics, and how these strategies connect to the obtained walking cycle (pros and cons). This would bring the reader to understand why AAN controllers may actually be the best solution available. I would suggest as recent literature on the topic: Effects of control strategies on gait in robot-assisted post-stroke lower limb rehabilitation: a systematic review and Current Trends in Robot-Assisted Upper-Limb Stroke Rehabilitation: Promoting Patient Engagement in Therapy
- “The feasibility of AGS and its assessment properties have been studied in two sessions with neurological patients [15].” Could you please complete the sentence with what was obtained within these sessions? The article must be self-contained without sending the reader to secondary articles.
Materials
- Line 134: All participants were used to train with the Lokomat à Consider removing, redundant.
Author Response
Zurich, 24th March 2023
Mrs. Della Diao
Dear Editor,
I wish to thank you for handling our manuscript entitled “Feasibility of an intelligent algorithm based on an Assist-As-Needed Controller for a Robot-Aided Gait Trainer (Lokomat) in Neurological Disorders: a longitudinal pilot-study” as well as we thank the reviewers for their constructive feedback.
Please find enclosed our detailed replies to the comments of the reviewers.
We hope you and the reviewers will find our revision satisfactory and the manuscript can now be accepted for publication on Brain Sciences.
Thank you again for your kind attention and assistance.
With best regards
Daniele Munari
RESPONSE TO REVIEWERS
We would like to express our appreciation for the time and effort put in by the editor and the reviewers. Their comments and suggestions have improved the quality of the manuscript.
Response to Reviewer #2
Comments to the Author
The authors investigate the feasibility of assist-as-needed controlling strategies (with the Lokomat device) in long-term neurorehabilitation. Despite a small cohort, the main finding of the article is that Adaptive Gait Support provides less force to the patient during the gait, fostering autonomous walking. The finding is important also given the relationship between support levels and walking tests values.
MAJOR
General
- Response to Reviewer #2 comment: Was LokomatPro involved in the study design/results choice and publishing?
- Authors’ reply: The LokomatPro was used as apparatus for the study. Two authors worked at the manufacturer of the LokomatPro, Hocoma, at the time of the study (see Conflict of interest statement). The other co-authors received no financial support from Hocoma and worked independently on this project.
Introduction
- Response to Reviewer #2 comment: I have a small doubt on the aim of the work. The authors are investigating feasibility of a new algorithm in long-term rehabilitation, already demonstrated in two short-term sessions. However, a feasibility study is a study where an assessment of the practicality of a proposed project plan or method is tested. However, I do not clearly see the difference of testing the feasibility of an algorithm used ‘long’ or ‘short’ term. What I mean is the following:
- The Lokomat has already been extensively validated for feasibility and also efficacy, both short- and long-term, thus the robot is not excessively problematic and it is usable in clinics.
- The algorithm AGS feasibility has already been demonstrated in short term.
Now my question, does using an algorithm for a long time change the request that the user needs to have from the algorithm (i.e., why do we need to evaluate its feasibility again?).
I would suggest the author to better describe what is the difference from previous studies either on Lokomat and on AGS, better detailing which are the differences between short- and long- term rehabilitation that require a second assessment of feasibility.
- Authors’ reply: Thanks for pointing out that the aim is not stated clearly. The Lokomat is used in clinical practice for 20 years, but the software has been recently updated with the addition of the AGS algorithm, which introduces a new automatic adaptation of the impedance of the controller. In this study we aimed to test only the feasibility of this particular new algorithm, not of the Lokomat per se. As the Reviewer correctly points out, this new approach has been already tested in another study [1], however there are three main differences between the current study and [1]: 1) the current study includes the final commercial version of the software, while [1] used a prototype research version , 2) the current study includes patients with different diagnosis, while [1] only included patients with SCI, 3) the current study explores the use of AGS in a real-world setting, during therapy extended over more weeks (for this reason we called it “long-term application”, while [1] only applied the AGS in 2 training sessions in a controlled environment.
We have also tried to explain these differences more clearly in the text.
Materials
Methods and protocols are clearly explained and easy to follow.
- Response to Reviewer #2 comment: Line 204: Please, check whether data is normal, report the result, then apply the proper statistical test for paired data (t-test or Wilcoxon)
- Authors’ reply: We would like to thank Reviewer #2 for the comment. The data is normal. In the section 2.6.1 is mentioned as follow “To test if the difference of the average robotic support level automatically determined by the AGS algorithm and manually adjusted by the therapist during conventional Lokomat training was significant, it was tested if the data was normally distributed, and a paired t-test was conducted. The paired t-test was used because differences in means were investigated. The normalized stiffness of the knee and the hip Lokomat was used to describe the robotic support level.”
- Response to Reviewer #2 comment: Line 210: I understand that 12 variables and 10 subjects are not easy to handle in multivariate analysis. However, I would suggest to avoid any sort of multivariate variable selection with only 10 subjects. Such analysis, and in particular backward and forward regressions are not only sub-optimal, but often misleading. This is the case when two or more independent variables are strongly dependent and the collinearity hypothesis is not respected (most likely with 12 gait-related variables). However, I understand the author’s necessity to perform a multivariate ‘feature ranking’. I would suggest to use an optimized (ratio between L1 and L2 and regularization strength) Elastic-Net linear regression and use the coefficients/p-values of that regression, much more robust to collinearity..
Authors’ reply: We understand the concerns of the reviewer and we thank him/her for the valuable suggestion. To check if the Knee stiffness at initial swing is indeed the most suitable variable to predict the walking tests, we applied an Elastic Net regression using the function lasso in MATLAB (which can be set to perform an Elastic net regression by tuning the value of alpha). Since cross-validation is used to determine the best regularization coefficient, lasso returns different results every time. To check if the results were consistent, we ran lasso 100 times. We then looked at the selection frequency of each variable: the most frequently selected variable was the same selected by the forward selection method. We added this explanation in methods and results.
- Response to Reviewer #2 comment - Line 213: Mixed effects linear models often require very high data cardinality. Did you perform an estimate of statistical power of these results? I understand the character of the analysis as exploratory and avoiding FDR correction. However, mixed effects introduce a third dimension in the analysis, thus requiring a much larger sample size. For what concern this analysis, I would suggest either removing or providing an estimate of effect size..
- Authors’ reply: We agree with the reviewer’s comment. Since we are limited by the sample size, we decided to remove this analysis.
Results
The participants are clearly described and I congratulate the authors for the patient-specific description of the enrollment phase.
Authors’ reply: Thank you for the comment, it is much appreciated.
- Response to Reviewer #2 comment: Line 289-290: From here I understand that the forward analysis has been performed on a dataset with some variables present for 10 patients while some other only for 6. This, often results in a list-wise elimination of subjects i.e., if one patient does not have a value for one variable, the whole patient is removed from the analysis, thus decreasing the cohort from 10 to 6 patients. If this is correct, the point raised at line 210 becomes even more crucial. With 6 patients I would avoid in any case multivariate analysis, and would only report univariate linear regressions between instrumental variables and outcomes.
- Authors’ reply: Correct, since the aim of the analysis described in section 2.6.2 was to find the best predictor(s) of walking ability (as measured with the 10MWT and TUG), we could not include data of four subjects who could not perform these tests. We did not want to perform a multivariate analysis given the risk of overfitting. We used the forward selection approach merely to identify which variable was able to best predict the walking tests (the choice of the Knee stiffness at initial swing was further confirmed by applying the Elastic Net approach, as suggested). Later, we only used this variable for all further analyses (reliability, display of longitudinal data within-patient). We did, in fact, only perform a univariate linear regression using Knee stiffness at initial swing as predictor, after this variable was chosen.
- Response to Reviewer #2 comment: Figure 7 is a fundamental result of the analysis, I would give the emphasis of the work on this concept, more than on the feasibility of the algorithm.
- Authors’ reply: Thank you for pointing this out. We have included this result in the abstract and added one sentence in the discussion, where the result was already extensively commented. We are unsure, however, if the limited number of subjects we have included allows us to draw strong conclusions on this point. We think that the results displayed in Figure 7 are for sure interesting and they hint that the relationship between isometric force and AGS data should be investigated more, but we refrain from making strong claims about it, without collecting further data. We find anyway very interesting that this analysis highlights some interesting facts about patients who could not walk, who had to be excluded from the other analyses: isometric force and AGS data could provide a measure of their “walking-related” function, even before they could actually walk overground. This is very relevant for the target groups who trains with the Lokomat (patients with severe gait disabilities), as they have few possibilities to access objective instrumented assessments and rely on ordinal, coarse scores, such as the FAC.
MINOR
Introduction:
- Response to Reviewer #2 comment: Even if clear to me, I may suggest to provide around line 40-45 a quick introduction on different control strategies (Force, Position, etc…) in lower-limb robotics, and how these strategies connect to the obtained walking cycle (pros and cons). This would bring the reader to understand why AAN controllers may actually be the best solution available. I would suggest as recent literature on the topic: Effects of control strategies on gait in robot-assisted post-stroke lower limb rehabilitation: a systematic review and Current Trends in Robot-Assisted Upper-Limb Stroke Rehabilitation: Promoting Patient Engagement in Therapy.
- Authors’ reply: Thank you for bringing these very relevant papers to our attention. We added a brief explanation of force, position and impedance control strategies in the introduction, also based on the papers you suggested, that have been cited, among others: ”In exoskeleton robots, control strategies provide support based generally on position control or on force control. In position control, the robotic joints are programmed to follow predefined trajectories; the impedance of the controller regulates how compliant they are to deviations. The relationship between deviation and applied force is regulated by the impedance, that can be fixed or adaptable (manually or automatically) [5, 6]. High impedance leads to a very constrained gait pattern, while low impedance increases compliance while reducing physical support to the movement. With force control, no pre-defined trajectory exists, and the control behavior is defined through reactions to specific actions by the patients, defined by events of gait as a cyclic and rhythmic process. This may result in more freedom of imposing one’s specific gait pattern, but, possibly, on a higher risk of abnormal trajectories, especially if the subject’s impairment is severe)[6-9].”
- Response to Reviewer #2 comment: “The feasibility of AGS and its assessment properties have been studied in two sessions with neurological patients [15].” Could you please complete the sentence with what was obtained within these sessions? The article must be self-contained without sending the reader to secondary articles.
- Authors’ reply: According to the recommendation of Reviewer #2, we added the following sentence “The feasibility of AGS and its assessment properties have been studied in two sessions with people after SCI [15]. In this study, fifteen participants with walking impairments and twelve able-bodied persons used the AGS during two Lokomat training sessions. The results showed that AGS can be used to quantify the support required by a patient while performing robotic gait training and is feasible in people with very different levels of gait impairment [1].”
Materials
- Response to Reviewer #2 comment: Line 134: All participants were used to train with the Lokomat à Consider removing, redundant..
- Authors’ reply: Thanks for the comment. According to the suggestion, we removed the sentence in the manuscript.
- Maggioni, S., et al., Assessing Walking Ability Using a Robotic Gait Trainer: Opportunities and Limitations of Assist-as-Needed Control in Spinal Cord Injury. PREPRINT (Version 1) available at Research Square, 2022.
- Perry, J. and J.M. Burnfield, Gait Analysis: Normal and Pathological Function. Vol. 2. 2010, Thorofare, New Jersey: Slack Inc.
- Baker, R. GAIT GRAPHS FOR BEGINNERS. 2013 [cited 2016; Available from: https://wwrichard.net/tag/gait-graphs/.
- Atkinson, G. and A.M. Nevill, Statistical methods for assessing measurement error (reliability) in variables relevant to sports medicine. Sports medicine, 1998. 26: p. 217-238.
- Marchal-Crespo, L. and D.J. Reinkensmeyer, Review of control strategies for robotic movement training after neurologic injury. J Neuroeng Rehabil, 2009. 6: p. 20.
- Campagnini, S., et al., Effects of control strategies on gait in robot-assisted post-stroke lower limb rehabilitation: a systematic review. J Neuroeng Rehabil, 2022. 19(1): p. 52.
- Riener, R., et al., Patient-Cooperative Strategies for Robot-Aided Treadmill Training: First Experimental Results. IEEE Trans Neural Syst Rehabil Eng, 2005. 13(3): p. 380-94.
- Blank, A.A., et al., Current Trends in Robot-Assisted Upper-Limb Stroke Rehabilitation: Promoting Patient Engagement in Therapy. Curr Phys Med Rehabil Rep, 2014. 2(3): p. 184-195.
- de Miguel-Fernández, J., et al., Control strategies used in lower limb exoskeletons for gait rehabilitation after brain injury: a systematic review and analysis of clinical effectiveness. Journal of NeuroEngineering and Rehabilitation, 2023. 20(1): p. 23.
Round 2
Reviewer 1 Report
Comments and Suggestions for Authors
Thanks to the authors for responding to all comments and revising the paper accordingly, thus increasing its overall quality.
Despite the authors' clarifications, the number of subjects involved in the study is really small to obtain significant and generalized results, especially considering that some participants could not fully adhere to the experimental protocol. The authors should have even discarded these subjects from the analysis. However, these weaknesses were correctly pointed out and highlighted in the limitations section, as well as the need to revise the experimental protocol in future studies. Since this is a feasibility and preliminary study on the use of the new AGS algorithm, as indicated by the authors, this necessary clarification can be considered sufficient: the results obtained could still provide valuable and helpful insights for other clinical studies based on the same exoskeleton.
Some suggestions regarding the revised version:
1) Figure 2: For clarity, the categories shown in the figure should be the same as those listed in subsection 2.5.1. In addition, it would be better to cite the study by Davis et al. [31] also in subsection 2.5.1.
2) Page 14: The new sentence “Table 4 shows the percentage of change between the first and the last AGS session.” probably refers to the wrong table (Table 4 was mentioned instead of Table 6). Also, it is not evident that the values in Table 6 refer to percentages. It would be better to add this information also in the caption of the figure.
Finally, concerning Item 17 (Round 1 comments), the authors did not provide an answer on the plot related to the P6 subject (“P6 shows the broader standard deviation in Robotic Support: this seems incongruent considering that P6 has the best FAC score (FAC=5). The same is true for the support set manually by the therapist (close to 100%), which is higher than that of P9 (less than 40%) with FAC=0”). For completeness, could the authors explain this point?
Author Response
RESPONSE TO REVIEWERS
We would like to express our appreciation for the time and effort put in by the editor and the reviewer. Comments and suggestions have improved the quality of the manuscript.
Response to Reviewer #1
Comments to the Author
Despite the authors' clarifications, the number of subjects involved in the study is really small to obtain significant and generalized results, especially considering that some participants could not fully adhere to the experimental protocol. The authors should have even discarded these subjects from the analysis. However, these weaknesses were correctly pointed out and highlighted in the limitations section, as well as the need to revise the experimental protocol in future studies. Since this is a feasibility and preliminary study on the use of the new AGS algorithm, as indicated by the authors, this necessary clarification can be considered sufficient: the results obtained could still provide valuable and helpful insights for other clinical studies based on the same exoskeleton.
Some suggestions regarding the revised version:
1) Figure 2: For clarity, the categories shown in the figure should be the same as those listed in subsection 2.5.1. In addition, it would be better to cite the study by Davis et al. [31] also in subsection 2.5.1.
- Authors’ reply: According to the suggestion of Reviewer #1, we implemented the changes in subsection 2.5.1 as follow “The determinants are (a) enjoyment, (b) anticipated used, (c) perceived usefulness, (d) perceived ease of use and (e) overall evaluation [31].”
2) Page 14: The new sentence “Table 4 shows the percentage of change between the first and the last AGS session.” probably refers to the wrong table (Table 4 was mentioned instead of Table 6). Also, it is not evident that the values in Table 6 refer to percentages. It would be better to add this information also in the caption of the figure.
- Authors’ reply: We apologize for the typo and according to the suggestion of Reviewer #, we corrected the sentence, added the percentage values in Table 6 and in the figure’s caption.
3) Finally, concerning Item 17 (Round 1 comments), the authors did not provide an answer on the plot related to the P6 subject (“P6 shows the broader standard deviation in Robotic Support: this seems incongruent considering that P6 has the best FAC score (FAC=5). The same is true for the support set manually by the therapist (close to 100%), which is higher than that of P9 (less than 40%) with FAC=0”). For completeness, could the authors explain this point?
- Authors’ reply: We thank Reviewer #1 for her/his detailed revision, and we would like to clarify this point. As described in the paper, we included different types of neurological diseases and of severity in order to understand how the algorithm responds during the robotic gait session in the Lokomat device. We noticed in our experience that patients with a lower severity level (e.g. FAC = 5 as P6) are more prone to impose their own (potentially unphysiological) gait pattern over the Lokomat gait pattern. In practice, as soon as the Lokomat support decreases, like it’s the case with the AGS, the patients with higher residual activity/strength/gait function have more possibility to deviate from the reference trajectory and sometimes they do so, and this could have been the case for P6. The high variability in robotic support displayed in Fig. 3 can also indicate that P6 tried different strategies while walking with AGS, leading to a broad range of resulting robotic support. On the contrary, patients with high impairment severity (e.g. FAC=0), can hardly deviate from the Lokomat reference trajectory, therefore they also require less robotic support. 40% of the Lokomat robotic support, if coupled with a high enough body weight support, could be sufficient to train patients who are passively following the robotic gait pattern.